# The *Hirudo Medicinalis* Microbiome Is a Source of New Antimicrobial Peptides

**DOI:** 10.3390/ijms21197141

**Published:** 2020-09-27

**Authors:** Ekaterina Grafskaia, Elizaveta Pavlova, Vladislav V. Babenko, Ivan Latsis, Maja Malakhova, Victoria Lavrenova, Pavel Bashkirov, Dmitrii Belousov, Dmitry Klinov, Vassili Lazarev

**Affiliations:** 1Federal Research and Clinical Center of Physical Chemical Medicine of Federal Medical Biological Agency, Moscow 119435, Russia; lizapavlova6@gmail.com (E.P.); daniorerio34@gmail.com (V.V.B.); lacis.ivan@gmail.com (I.L.); maja_m@mail.ru (M.M.); lavrv1995@gmail.com (V.L.); pavel.bashkirov@niifhm.ru (P.B.); klinov.dmitry@mail.ru (D.K.); 2Moscow Institute of Physics and Technology, 141700 Dolgoprudny, Moscow 141701, Russia; 3Department of biochemistry, Faculty of Biology, Lomonosov Moscow State University, Moscow 119991, Russia; 4Sechenov First Moscow State Medical University Sechenov University, Moscow 119991, Russia; belik2011@mail.ru

**Keywords:** microbiome, metagenome, *Hirudo medicinalis*, antimicrobial peptides, bioinformatic analysis, secondary structure, circular dichroism, α–helix

## Abstract

Antimicrobial peptides (AMPs) are considered a promising new class of anti-infectious agents. This study reports new antimicrobial peptides derived from the *Hirudo medicinalis* microbiome identified by a computational analysis method applied to the *H. medicinalis* metagenome. The identified AMPs possess a strong antimicrobial activity against Gram-positive and Gram-negative bacteria (MIC range: 5.3 to 22.4 μM), including *Staphylococcus haemolyticus*, an opportunistic coagulase–negative pathogen. The secondary structure analysis of peptides via CD spectroscopy showed that all the AMPs except pept_352 have mostly disordered structures that do not change under different conditions. For peptide pept_352, the α–helical content increases in the membrane environment. The examination of the mechanism of action of peptides suggests that peptide pept_352 exhibits a direct membranolytic activity. Furthermore, the cytotoxicity assay demonstrated that the nontoxic peptide pept_1545 is a promising candidate for drug development. Overall, the analysis method implemented in the study may serve as an effective tool for the identification of new AMPs.

## 1. Introduction

The massive uncontrolled use of antibiotics has led to the emergence and widespread distribution of pathogenic microorganisms. The need to develop new strategies for combating resistant microorganisms requires the use of effective therapeutic drugs that differ in their mechanism of action from the currently used anti-infectious agents. Natural compounds could serve as an alternative to synthetic antibiotics. For example, antimicrobial peptides (AMPs) are universal and evolutionarily ancient components of the innate immune system and are a part of the first line of host defence against invading pathogens [1]. AMPs are capable of inhibiting the growth of various microbes and have an advantage compared to traditional antibiotics. The unique mechanism of action of AMPs is to disrupt the integrity of the bacterial cell membrane, which leads to cell death [2,3]. The possibility of developing resistance to AMPs is limited since the pathogen requires structural and electrophysiological changes in the cell membrane to counteract the peptide.

To date, thousands of antimicrobial peptides have been identified [4,5]. AMPs are very diverse in their sources, structures, and sequence features. The most widespread AMPs are short amphipathic α–helical peptides [6]. Many AMPs are currently being examined during preclinical and clinical trials as antimicrobials and anti-inflammatories for therapeutic applications. Currently, only several peptide-derived treatments have come to the market. For example, PAC–113 (P–113), a peptide derived from a naturally occurring histatin, is used as a topical treatment for oral candidiasis infections [7]. The semisynthetic lipoglycopeptide dalbavancin has been approved for the intravenous treatment of acute skin infections caused by Gram-positive microorganisms [8]. Although the majority of AMPs failed in studies due to toxicity and instability, the number of new compounds in preclinical trials has grown rapidly. The development of new AMPs is aimed at maximizing the antimicrobial activity while minimizing toxicity. The central methodologies to design improved AMPs are site-directed mutagenesis, computational design approaches, synthetic library analysis, and template-assisted methodologies [9]. However, different organisms remain the main source of new AMPs, and high-throughput screening of omics data is widely used to identify new compounds [10,11].

The medicinal leech has an extensive repertoire of biologically active proteins that evolved due to extreme living conditions and hematophagous ways of life [12,13]. The microbiome of *H. medicinalis* plays a crucial role in host physiology and health and is tightly controlled by the medicinal leech or the microbiome itself [14]. A previous study showed that antibiotics, even at concentrations below the clinical breakpoint, cause dramatic changes in microbial populations and lead to a marked increase in the number of resistant strains [15]. The commensal bacteria both protect the gut against microbial colonization and must be resistant to the antimicrobials of medicinal leech presented normally in the absence of inflammation. The diversity of the microbiota provides the host with unique compounds of immune defence, such as antimicrobial peptides. Microbiome analysis of *H. medicinalis* will provide a better understanding of the mechanism underlying medicinal leech and symbiotic microbe interactions. Recently, we annotated the *H. medicinalis* genome and described the identification of novel anticoagulant proteins [16]. Moreover, we developed an in silico screening method that allowed the identification of new AMPs with a low haemolytic and toxic activity. The present study describes the first implementation of this strategy in the *H. medicinalis* metagenome. We identified new AMPs and examined and compared the mode of action of peptides and their effects against Gram-positive and Gram-negative bacteria, including the opportunistic pathogens *S. haemolyticus* and *S. aureus.*

## 2. Results

### 2.1. Metagenome-Wide Search for Antimicrobial Peptides

The contigs of the *H. medicinalis* microbiome were generated during a previous study involving the *H. medicinalis* genome analysis due to the binning procedure [16]. De novo annotation of the metagenome *H. medicinalis* allowed us to generate a database of proteins that can serve as a source of AMPs. The predominant bacterial taxa of the microbiome included *Agrobacterium*, *Aminobacter*, *Bradyrhizobiaceae*, *Chitinophagia*, *Myxococcales,* and *Sphingobacteriia*. The overall metagenome protein database included 24,987 complete amino acid sequences. The majority of the proteins belong to the *Aminobacter* taxa. At the initial step of the bioinformatic analysis, we filtered sequences that may exhibit an antimicrobial activity using the DBAASP v.2 server [17]. This application uses not only the physicochemical properties of the peptides from the database but also the chemical structures of AMPs and the empirically determined activities of peptides against different target microbes for antimicrobial propensity prediction. In total, 472 proteins were defined as possible antimicrobial agents. A total of 213 putative antimicrobial compounds belong to *Chitinophagia* taxa. According to the blast analysis, the functions of these proteins are not determined. Next, we determined the precise sequence of putative AMPs using the AMPA server [18]. The web application allows fast identification of antimicrobial patterns in proteins. Therefore, we extracted 238 putative AMP sequences with lengths from 12 to 37 amino acid residues. For all peptides, the antimicrobial potential was calculated by ADAM, CAMP–R3, iAMP–2L, and AmPEP predictors [5,19,20,21]. We filtered 65 peptides with the highest antimicrobial propensity scores. As the majority of AMPs are cationic positively charged peptides prone to form an α–helix, the secondary structures of potential candidates were analyzed using the I–TASSER–MR server [22]. The server constructed the 3D models of peptides according to the amino acid sequence. According to the secondary structure prediction among 65 filtered peptides with the highest antimicrobial propensity scores, the predicted secondary structures were only α–helical or unstructured conformations. From the 65 peptides, we chose five peptides with the highest antimicrobial propensity scores, which can exhibit the α–helical structure and additionally one peptide with unstructured conformation and high propensity score according to the predictions (Table 1).

### 2.2. Secondary Structures of AMPs

According to the secondary structure predictions, all peptides except peptide pept_1303 could form an α–helix, whereas peptide pept_1303 was unstructured (Appendix A). The deconvolution analysis of CD spectra demonstrated that AMPs in aqueous solutions had a small β–strand content (<30%) and were mainly disordered (>50%) (Figure 1, Appendix A). Under buffer conditions, the secondary structure of peptides slightly changed but remained mostly the same. The growth of α–helical content in the buffer compared to the water solution was observed only for peptide pept_352, indicating a transition from the disordered coil to the helix fold. In general, AMPs are known to form random coil structures in physiological solutions, but the secondary structure of peptides can change under the membrane environment [23]. We analyzed the structural modulation of peptides upon interaction with large liposomes prepared from electrically neutral phosphatidylcholine (POPC). Upon interaction with phospholipids, the structures of almost all studied peptides did not vary from the structure under buffer conditions. An exception was the peptide pept_352, which had an increased α–helical content in the membrane environment, indicating that the peptide pept_352 adopts an α–helical conformation and consequently is embedded in the bacterial membrane. CD spectra of melittin, a membrane-lytic α–helical peptide, were analyzed as a positive control (Figure 1, Appendix A). Previous CD spectroscopy experiments examining the melittin secondary structure determined that melittin is mostly unstructured in aqueous solution but adopts an α–helical conformation in the presence of lipid vesicles [24]. Moreover, the degree of the transition from a completely unstructured fold to a predominantly helical structure is proportional to the POPC liposome concentration [25].

### 2.3. Antimicrobial Activity of AMPs

The antimicrobial activity of the six putative AMPs against Gram-positive and Gram-negative bacteria was evaluated. As a positive control, the well-known AMP melittin was used. All synthesized peptides exhibited a broad spectrum of antimicrobial activity (Table 2). Peptide pept_352 had potent antimicrobial activity against all bacteria. It is worth mentioning that pept_352 kills MRSA *S. aureus* ST 88, a major skin pathogen, at a concentration of 19.9 μM [26]. Peptides pept_1303, pept_1545, and pept_84 also showed potent antimicrobial activity against Gram-positive and Gram-negative bacteria of the same order of magnitude concentrations. Peptide pept_970 was active against *E. coli* MG1655 and *B. subtilis* 168HT at MICs equal to 79 and 19.8 μM, respectively. Furthermore, peptide pept_148 was potent against only *B. subtilis* 168HT. These results show that our metagenome-wide screening method can be applied for designing novel AMPs.

### 2.4. Cytotoxic Assays of AMPs

The cytotoxicity of AMPs was investigated by viability assays of McCoy cells treated with peptides for 24 and 48 h at a concentration equal to 4x MIC. We applied two techniques: Fluorescence imaging of dead and living cells (live/dead viability/cytotoxicity assay, Table 3, Appendix A) and a colorimetric assay (lactate dehydrogenase assay, Appendix A). In reliance on the results of both approaches, we determined that peptides pept_352 and pept_970 had a strong cytotoxicity comparable to melittin. Peptides pept_84 and pept_1303 reduced the cell viability to a lesser but still severe degree. Cells exposed to peptides pept_1545 and pept_148 did not exert any inhibition of cell growth even after 48 h.

### 2.5. AMP Activity by Propidium Iodide Penetration Assay

The viability of *E. coli* MG1655 and *B. subtilis* 168HT after 1 h of peptide treatment was assessed by fluorescence microscopy (Appendix A, Appendix A). After peptide incubation, cells were stained with PI, the red fluorescent agent propidium iodide (PI), which penetrates only dead bacteria and intercalates with intracellular DNA. The increase in the fluorescence signal in comparison with nontreated live cells demonstrated damage to the bacterial cell and eventually the death of the bacteria. The rate of peptide activity on *B. subtilis* 168HT after the addition of the AMPs was evaluated by the kinetics of PI penetration (Figure 2). All peptides were able to kill bacteria as indicated by the increase in the fluorescence signal of the PI dye compared to the negative control. Moreover, the fluorescence detected at the primary stage of incubation illustrated immediate damage to the cells at the moment that peptides were added. This suggests the rapid mechanism of action of the analyzed AMPs.

### 2.6. Bacterial Cell Morphological Changes after AMPs Exposure

The effect of AMPs on bacterial surface morphology after incubation with peptides was analyzed by scanning electron microscopy (SEM) (Figure 3, Appendix A). As shown in Figure 3, untreated bacteria were numerous and had predominantly smooth surfaces without defects. The bacterial treatment with AMPs caused dramatic cellular damage, with the formation of cellular debris. Beyond that, a marked reduction of the cells was observed, although a few intact cells without morphological differences were noticed in almost every sampling. The treatment of bacteria with the AMPs at a concentration of 1/2x MIC did not cause significant damage to the cells, although debris was also found. Overall, the AMP effect was comparable to that of the membranolytic AMP melittin. For *B. subtilis* 168HT treated with peptides (especially with peptide pept_84), a significant increase in cell size was observed when the damaged bacterial aggregates were still present. This observation can be attributed to the fact that peptides can inhibit cell division.

## 3. Discussion

Increased bacterial resistance to conventional antibiotics requires the development of new therapeutic agents. Antimicrobial peptides, compounds that can combat infections and have an advantage over commonly used antibiotics, are regarded as promising molecules to effectively kill microorganisms [27,28]. During recent decades, a range of strategies have been developed to design new peptides [29,30,31]. These entirely novel antimicrobial peptides can be developed by high-throughput screening of libraries of randomly synthesized amino acid sequences [10]. Another approach is the usage of in silico methods to identify functionally active proteins that peptide-derived could exhibit the antimicrobial activity. Specifically, the in vivo cleavage products of proteins such as histones, lactoferrin, and buforin exert potent antimicrobial activities [32,33]. Previously, we implemented a computational algorithm for the *H. medicinalis* genome assembly analysis to identify novel AMPs derived from proteins of the medicinal leech [12]. The developed peptides had broad antimicrobial activity and exhibited low toxicity and haemolytic effects. The present study was focused on the *H. medicinalis* microbiome analysis.

The microbial community of the medicinal leech is involved in vital processes of the worm, such as digestion, protection from pathogens, and regulation of immunity [13,34]. Metagenomic analysis of microbiota allows not only the identification of bacteria and interactions with organisms but also the specification of groups of genes encoding functionally active proteins, especially those that have antimicrobial effects. In this study, we applied an in silico method to examine the metagenome of *H. medicinalis* for the identification of protein-derived putative AMPs. For the data analysis, we applied several online available algorithms. At the initial step of the research, we discerned amino acid sequences of *H. medicinalis* microbiome proteins that are capable of exerting the antimicrobial activity. Next, we defined the amino acid sequences accountable for the antibacterial effect. As a result, we selected six putative AMPs.

The selected peptides belong to the two phylums of Gram-negative bacteria *Bacteroidetes* (*Sphingobacteriia* and *Chitinophagia*) and *Proteobacteria* (*Aminobacter*), bacteria which are typically members of host-associated microbial communities. The literature data concern only investigations of metagenomic data and individual genomes from soil samples. There is little information on AMPs identified in these bacteria. At the moment, only non-ribosomal antimicrobial peptides isolated from marine Proteobacteria has been known [35]. According to the existing observations and studies on the metagenomes of leeches of the genus *Hirudo* these taxa of microorganisms are common symbionts of leeches. The genus *Aminobacter* belongs to the order *Rhizobiales*, representatives of this order are components of the microbiota from the mucus and skin of the leech [36,37]. Representatives of bacteria of the class *Sphingobacteriia* and *Chitinophagia* are associated with metagenomes of the skin and excretory system of leeches [37,38]. For example, bacteria from the genus *Niabella* (Class *Chitinophagia*) can be found in leech cocoons, during the development of the leech, these bacteria colonize the bladder of the leech [39]. In some cases, representatives of these classes of bacteria were observed in insignificant quantities in various parts of the leech’s digestive system [40,41]. Thus, it can be assumed that these bacteria are endosymbionts and use antimicrobial peptides to compete successfully in the process of colonization of certain leech organs.

In general, natural AMPs are amphipathic cationic peptides that differ significantly in sequence and structure. The secondary structure predictions of selected peptides were accomplished via the I–TASSER–MR server [24]. According to the predictions, only peptide pept_1303 adopts an unstructured fold, whereas other peptides indeed form α–helices. The secondary structure prediction of peptides is a difficult task due to the small size of the compounds. Moreover, predictors did not take into account the fact that peptides change the conformation under different conditions. Therefore, the AMP structure evaluation is possible only by an experimental approach. To establish the secondary structure of the putative AMPs, we used CD spectroscopy. The analysis of CD spectra demonstrated that peptide pept_352 tends to adopt an α–helical conformation in the buffer and in the presence of POPC liposomes. Similarly, α–helical melittin changes from predominantly unstructured in aqueous solution to α–helical folds after the addition of POPC liposomes [27]. The other peptides remained mainly unstructured under any condition. Previously, it was demonstrated that the mechanism of action of AMPs is dependent on the surface charge and hydrophobicity of liposomes of mixed phospholipid compositions [42]. The secondary structure of the peptide changes under interactions with bacteria and can vary with different lipid membrane compositions. It is, therefore, possible to determine the exact mechanism of action of AMPs by studying the change in structural conformation under different conditions.

Exploration of the biological activity of chemically synthesized peptides revealed that all peptides exert an antimicrobial activity. The most drastic effect was demonstrated by peptide pept_352, as it killed bacteria at a relatively small MIC equal to 10 μM against *E. coli* K12 substr. MG1655, *B. subtilis* 168HT, *S. haemolyticus* 527, and 19.9 μM against *S. aureus* ST 88. Other peptides pept_1303, pept_84, and pept_1545 also showed potent antimicrobial activity. We suppose that the difference between the response of *S. haemolyticus* and *S. aureus* to peptides was dependent of the membrane lipid composition of bacteria. The considerable membrane plasticity of *S. aureus* provides antimicrobial resistance to peptides [43,44]. Nonetheless, cytotoxic assays of the AMPs demonstrated that peptides pept_1545 and pept_148 did not cause cell death at high concentrations after 24 and 48 h incubation. The main disadvantage of AMPs is their high toxicity, which prevents their use in practical treatment. For example, melittin, a bee venom peptide, exerts excellent antimicrobial activity but possesses nonspecific cytotoxicity, and haemolytic activity has prevented its therapeutic applications [45]. There exist several approaches to reduce the cytotoxic effect of AMPs, such as amino acid substitution, resizing of the length of peptides, and the design of hybrid peptides [46,47]. A further experimental investigation is needed to test which modification of the described AMPs can change their cytotoxicity.

AMPs are known to exhibit broad spectrum antimicrobial effects on pathogens [2,48]. According to numerous studies, the actions of AMPs on bacteria are diverse and appear to be dependent on the target and the peptide concentration. The speed of the antimicrobial effect of different AMPs can vary greatly. For example, Porto et al. showed that pore-forming melittin kills *E. coli* ATCC25922 during 10 s, while the lytic effect of guavanin 2 took 10 min [32]. The fluorescent dye uptake assay demonstrated that all tested peptides caused the death of *B. subtilis* 168HT in a relatively fast manner (in the first 50 s). Therefore, the speed of the lytic effect of the identified AMPs is comparable to that of melittin.

The action of AMPs on bacteria was confirmed by the SEM of bacteria treated with peptides. AMPs caused dramatic inhibition of bacterial growth as well as the formation of cellular debris in the AMP-treated samples. It is important to highlight that the *B. subtilis* 168HT treatment with peptides caused a dramatic cell size gain when damaged bacterial aggregates were also observed. This observation attests to the fact that peptides can inhibit cell division. The specific method for inhibiting bacterial growth of the identified AMPs remains to be determined. A further study of the properties of AMPs and understanding the mechanism of action of such compounds will allow us to further understand not only the mechanisms of the innate immune system but also facilitate the development of new therapeutic agents. According to the literature data, α–helical AMPs are characterized by a lytic mechanism of action via direct integration into the membrane. Peptide pept_352 adopts an α–helical conformation in the presence of POPC liposomes, while it is unstructured in aqueous solution. Altogether, the research results suggest that peptide pept_352 may act as a membranolytic agent and cause cell death by disturbing the cell membrane structure. The mechanism of action of other identified AMPs is obscure; nonetheless, they do affect the bacterial cell wall, causing bacterial death.

## 4. Materials and Methods

### 4.1. Data Set Analysis

The metagenome contigs were retrieved previously in the process of *H. medicinalis* genome annotation [16]. Bacterial contig groups were annotated using PROKKA (University of Melbourne, Melbourne, Australia) and GhostKOALA (Kyoto University Bioinformatics Center, Kyoto, Japan) software [49,50]. Amino acid sequences with antimicrobial properties among de novo assembled microbiome proteins were retrieved using the DBAASP v2. Server (IBCEB, Georgia; NIAID, USA) [17]. The AMPA (Center for Genomic Regulation of Barcelona, Barcelona, Spain). server with the default setup was used to select the putative AMPs [20]. The antimicrobial potential of peptides was estimated by using the available online algorithms ADAM (National Taiwan Ocean University, Keelung City, Taiwan), CAMP–R3 (Biomedical Informatics Center, Mumbai, India), iAMP–2L (Jingdezhen Ceramic Institute, Jiangxi, China), and AmPEP (University of Macau, Macau, China) [5,21,22,23]. In the following analysis, peptides with the highest scores remained. The secondary structures of potential candidates were analyzed using the I–TASSER–MR (University of Michigan, Ann Arbor, MI, USA) server [22]. As a result, we chose five peptides with the highest antimicrobial propensity scores, which can exhibit the α–helical structure and additionally one peptide with an unstructured conformation and high propensity score according to the predictions. The physicochemical properties of peptides were calculated using an APD3 (University of Nebraska Medical Center, Omaha, NE, USA) server for the amino acid composition, hydrophobic ratio, Boman index, and molecular weight [4] (Appendix A). 

### 4.2. Peptide Synthesis

The peptides were synthesized by the N–9–fluorenylmethyloxycarbonyl (Fmoc) strategy using a Liberty Blue automated microwave peptide synthesizer (CEM, Stallings, NC, USA), as previously described [12]. The purification of the peptides was accomplished by liquid chromatography using an AKTA pure chromatography system (GE Healthcare, Chicago, Illinois, USA) with purity >95%. The sequence and degree of purity (>95%) were confirmed by mass spectrometry by an ULTRAFLEX MALDI–TOF/TOF mass spectrometer (Bruker, Fremont, CA, USA).

### 4.3. Antimicrobial Activity Assay

The minimum inhibitory concentration (MIC) of peptides was determined by the standard microtiter dilution method, as performed previously [12,51]. Briefly, the MIC was determined by growing the microorganisms in 96-well microtitration plates in the presence of a two-fold serial dilution of the peptides. The MIC was determined as the lowest concentration of the peptide that completely inhibits the growth of the microorganism, similar to the negative control with pure broth without bacteria. Mueller Hinton broth (MHB, BD Difco, Thermo Fisher Scientific Inc., Waltham, MA, USA) was used for all cultures, including Bacillus subtilis 168HT, Staphylococcus aureus ST 88, Staphylococcus haemolyticus 527 (clinical isolate), and Escherichia coli K12 substr. MG1655 strains. Each experiment was performed in triplicate with positive (melittin (Merck Company Inc., Kenilworth, NJ, USA)) and negative (without peptide) inhibition controls.

### 4.4. Circular Dichroism Spectroscopy

Circular dichroism (CD) spectra were recorded on a Chirascan spectrophotometer (Applied Photophysics, Leatherhead, UK), equipped with a thermostated cuvette holder. Measurements were recorded at 20 °C and performed in quartz cuvettes of 0.5 mm path length between 190 and 260 nm at 1 nm intervals. CD spectra were obtained in the following conditions: Aqueous solution (pH 7), buffer (100 mM KCl, 1 mM HEPES, 0.2 mM EDTA, pH 7.4) and buffer with the addition of POPC liposomes to the final liposome concentration equal to 0.2 g/l POPC liposomes were prepared by extrusion through polycarbonate filters with 100 nm pore diameters using a Mini-Extruder (Avanti Polar Lipids Inc, Alabaster, AL, USA), as described previously [12]. The peptide concentration was 250 μM for all experimental conditions. The relative secondary structure content was calculated from the ellipticity values at 200 nm and above [52]. CD data were processed with the BestSel ( ELTE Eötvös Loránd University, Budapest, Hungary) method [53].

### 4.5. Viability of Bacteria by PI Penetration

The viability of *B. subtilis* 168HT and *E. coli* K12 substr. MG1655 after treatment with antimicrobial peptides was visualized by fluorescence microscopy. Exponentially growing bacteria (1 × 107 cfu/mL) were incubated with peptides at a final concentration equal to 2× MIC at 37 °C for 1 h. After incubation, the bacterial suspension was stained with propidium iodide (PI (Thermo Fisher Scientific Inc., Waltham, MA, USA)) at a final concentration of 20 μg/mL for 15 min in the dark. Nonviable bacteria were investigated using a Nikon Eclipse Ti fluorescence microscope (Nikon, Japan) with a Plan Fluor 60×/0.70 objective (Nikon, Tokyo, Japan) with a set of filters providing excitation/emission (528–553 nm/590–650 nm) for PI. The images were captured with an ORCA–Flash4.0 camera (Hamamatsu Photonics, Hamamatsu, Japan) with a 2 s exposure time. Fluorescent images were processed using the FIJI software to determine the image intensity histogram (fluorescent intensity) [54].

In addition, the kinetics of PI penetration over 5 min was determined by spectrofluorometry for *B. subtilis*. Exponentially growing bacteria (1 × 10^8^ cfu/mL) were resuspended in PBS after centrifugation (1000× *g*, 10 min, 4 °C) and were stained with PI at a final concentration of 50 μM for 30 min in the dark. After the addition of peptides (final concentration two-fold above the MIC), with a final volume of 350 μL, the fluorescence intensity of PI was measured at a maximum of 650 nm (excitation at 490) using a spectrofluorometer FluoroMax Plus (Horiba Scientific, Tokyo, Japan). The results are represented with the corresponding control experiment (PBS addition).

### 4.6. Cytotoxic and Cell Viability Assays

The cytotoxicity of peptides towards the mouse fibroblast cell line McCoy was determined via the LIVE/DEAD™ Viability/Cytotoxicity Kit (Thermo Fisher Scientific Inc., Waltham, MA, USA) according to the manufacturer′s recommendations, as performed previously [12,45]. Briefly, 100 μL of McCoy cells per well were seeded on 96-well culture plates at a density of 2 × 10^5^ cells/mL and incubated for 24 and 48 h at 37 °C with peptides at a final concentration equal to 4× MIC. Melittin was used as a negative control, and McCoy cells treated without peptides were used as a positive control. After peptide exposure, cells were washed and then stained with PBS containing calcein AM and ethidium homodimer–1 at concentrations of 0.3 and 1.25 μM, respectively, for 30 min at 37 °C. After staining, the cells were washed twice with PBS. The images were captured using an epifluorescence microscope with the respective filter cubes from 10 occasionally selected fields of view captured for each well with a 20× objective. Images were captured for FITC and TRITC channels. The numbers of cells in both channels were quantified with the ImageJ software (version 1.48; RSB).

The cytotoxicity of peptides was also measured by a lactate dehydrogenase (LDH) assay kit (Merck Company Inc., Kenilworth, NJ, USA) according to the manufacturer′s recommendations. Briefly, 100 μL of McCoy cells per well were seeded on 96-well culture plates at a density of 2 × 10^5^ cells/mL and incubated for 24 h at 37 °C with peptides at a final concentration equal to 4× MIC. Melittin was used as a negative control, and McCoy cells treated without peptides were used as a positive control. The growth medium without peptides was used as blank absorbance readings. After the peptide treatment, the plate with cells was centrifuged for 10 min at 900 g, and 50 μL of medium from each well was transferred to another plate. Fifty microlitres of the prepared LDH kit solution were added to each well. All manipulations were performed in the dark. The absorbance of the wells was measured after 15 min incubation in the dark at wavelengths of 495 and 690 nm using a Multiscan Ascent microtiter plate reader (Thermo Fisher Scientific Inc., Waltham, MA, USA). The specific absorbance of each sample was calculated according to the equation:Absorbance = A_495_ nm (Test) − A_495_ nm (Blank) − A_690_ nm (Test)

The percentage of cell viability was recalculated according to the control values.

### 4.7. Scanning Electron Microscopy

Bacterial strains in the mid-log growth phase were treated with the peptides (pept_84, pept_352, pept_1545, and pept_1303) at 37 °C for 8 h for *B. subtilis* 168HT, *E. coli* K12 substr. MG1655, and 24 h for *S. haemolyticus* 527. The bacterial suspension was deposited on a clean, sterile glass slide coverslip and was fixed using a 2.5% glutaraldehyde solution for overnight at 4 °C. Then, samples were dehydrated with increasing ethanol percentages (10%, 20%, 30%, 40%, 50%, 60%, 70%, 75%, 80%, 85%, 90%, 96% for 15 min in each solution). Then, the samples were incubated in a hexamethyldisilazane (HMDS, Reachem, Russia): ethanol mixture (1:1 by volume) for 10 min and in 100% HMDS overnight until complete HMDS evaporation [55]. After chemical drying, the samples were covered by a 10 nm gold-palladium alloy using Sputter Coater Q150T (Quorum Technologies, Lewes, UK). The samples were characterized by a Zeiss Merlin microscope equipped with GEMINI II Electron Optics (Zeiss, Oberkochen, Germany) at 4–8 kV accelerating voltage and 200–400 pA probe current. 

## 5. Conclusions

The present study reports the identification of new antimicrobial peptides derived from proteins of the *H. medicinalis* microbiome. Using different experimental techniques, we conducted a functional investigation of the identified AMPs. The peptide pept_1545 is a promising candidate for therapeutic agent development due to its broad antimicrobial activity and non-toxic effect on eukaryotic cells. Other peptides have a potent antimicrobial activity and toxicity. Nevertheless, the investigation of these peptides will clarify the antimicrobial activity–toxicity relationship that could promote the development of novel AMPs with low toxicity effects.

## Figures and Tables

**Figure 1 ijms-21-07141-f001:**
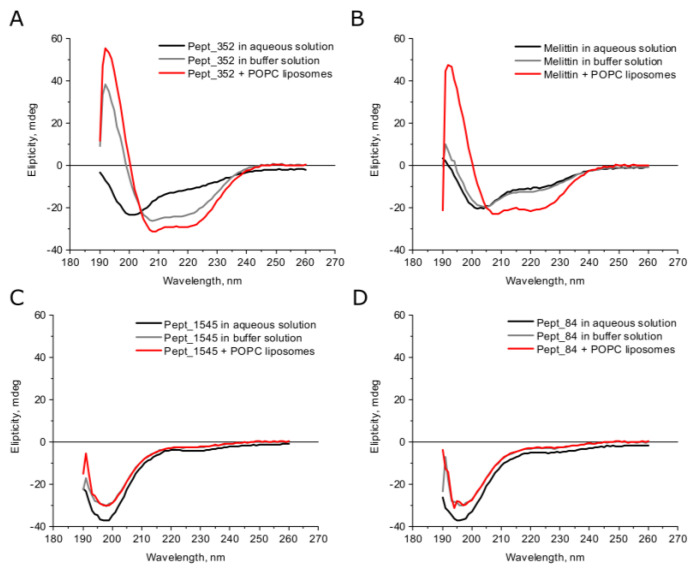
Secondary structure analysis of peptides under different conditions by CD spectroscopy. Circular dichroism (CD) spectra of peptides pept_352 (**A**), pept_1545 (**C**), pept_84 (**D**), and melittin (**B**). Buffer conditions: 100 mM KCl, 1 mM HEPES, 0.2 mM EDTA. The final concentration of the peptide in the experiments was 250 μM.

**Figure 2 ijms-21-07141-f002:**
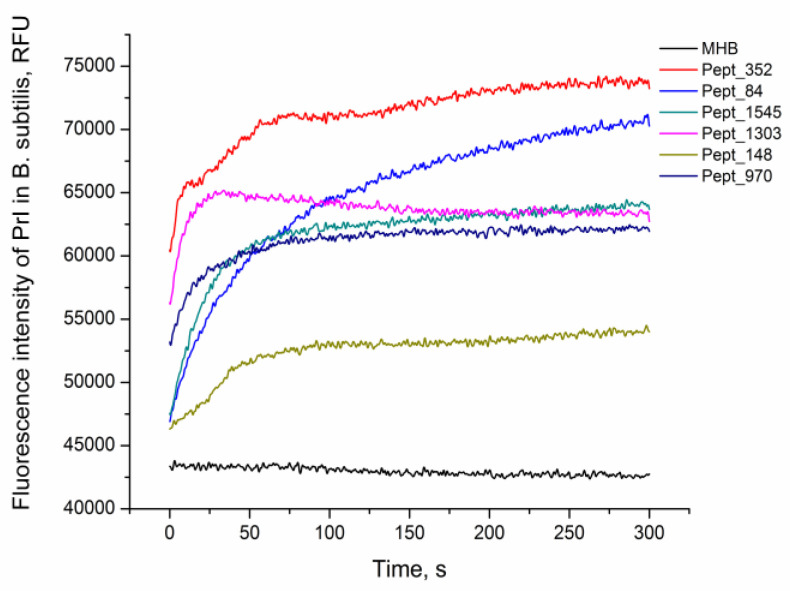
AMP activity against *B. subtilis*. Kinetics of propidium iodide (PI) penetration after the addition of AMPs to *B. subtilis* at a concentration equal to 2× MIC. The negative control corresponds to the bacteria incubated with PI without peptide (MHB).

**Figure 3 ijms-21-07141-f003:**
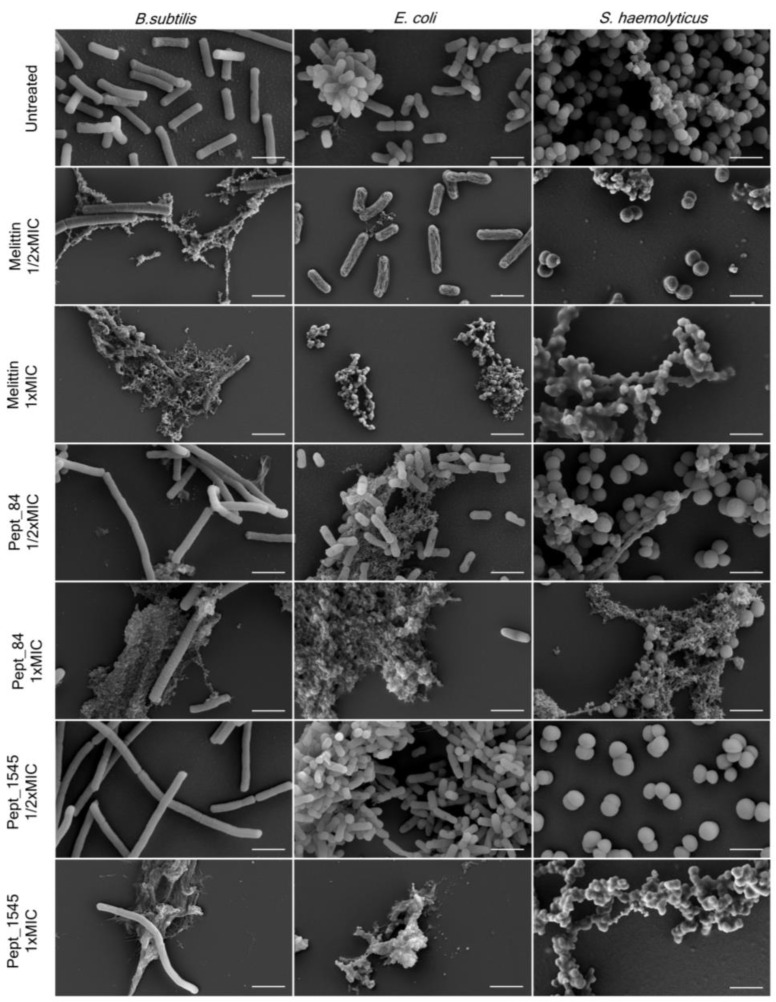
AMPs affect the bacterial membrane integrity. Scanning electron microscopy (SEM) images of bacterial strains treated with peptides pept_84 and pept_1545 and a positive control, melittin *B. subtilis* and *E. coli* were treated with peptides at a final concentration equal to 1/2× and 1× MIC for 8 h, and *S. haemolyticus* was treated with peptides at a final concentration equal to 1/2× and 1× MIC for 24 h. Scale bars: 2 μm.

**Table 1 ijms-21-07141-t001:** The list of antimicrobial peptides (AMPs) from the *H. medicinalis* microbiome.

Peptide	Organism	Amino Acid Sequence	Length, a. a.	Molecular Weight, Da
pept_1303	*Aminobacter*	IGRHFKRRNSIWGICWF	17	2176.58
pept_148	*Aminobacter*	VLIRGLIHMLRGG	13	1434.81
pept_970	*Sphingobacteriia*	FVKILAKLVNYAKN	14	1621
pept_1545	*Chitinophagia*	FLIGKAIKRKFCLRSVWNA	19	2250.78
pept_352	*Sphingobacteriia*	KKGKSFKQLHIIVHLVKSWLRTILTHI	27	3224.98
pept_84	*Chitinophagia*	IVKRFFRISYKLQSLKIIKGKRTFT	25	3071.79

**Table 2 ijms-21-07141-t002:** Antimicrobial properties of AMPs.

	MIC (μM)			
Peptide	*E. coli* K12 substr MG1655	*B. subtilis* 168HT	*S. aureus* ST 88	*S. haemolyticus* 527
pept_1303	14.8	14.8	>100	14.8
pept_148	>100	22.4	>100	>100
pept_970	79	19.8	>100	>100
pept_1545	14.3	7.2	>100	14.3
pept_352	10	10	19.9	10
pept_84	10.5	5.3	>100	5.3
Melittin	5.7	1.5	1.5	5.7

**Table 3 ijms-21-07141-t003:** Viability AMP-treated McCoy cells calculated from fluorescent images.

Peptide	Cell Viability, %
pept_1303	25.1 ± 8.3 *
pept_148	98.3 ± 0.9
pept_970	2.7 ± 2.1 **
pept_1545	98.7 ± 1.3
pept_352	0.2 ± 0.1 ***
pept_84	11.2 ± 7.4 *
Melittin	0.3 ± 0.1 ***

Cells were treated for 24 h with peptides at a final concentration equal to 4× MIC. The values are indicated as the mean ± standard deviation (SD) (*n* = 3). Statistically significant differences between the control and experimental groups were determined by the non-parametric Kruskal–Wallis test, * *p* < 0.05, ** *p* < 0.01, *** *p* < 0.005.

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
