# Peer review of "The Hirudo Medicinalis Microbiome Is a Source of New Antimicrobial Peptides"

_ijms, 2020, doi:10.3390/ijms21197141_

Round 1
Reviewer 1 Report
In the present paper, the authors analyze the antimicrobial characteristics of six different bacterial peptides, selected after a specific in silico screening analysis conducted on Hirudo medicinalis metagenome. The MIC (minimum inhibitory concentration) of every chosen peptide have been calculated observing the effects on different Gram-negative and Gram-positive bacterial strains and the relative antibacterial role was evaluated by different techniques. Moreover, the cytotoxicity of these amino acid sequences has been tested on McCoy cells. Taken together, the results of these experiments highlight as these putative AMPs show a different antimicrobial activity and cytotoxicity.
In my opinion, although the manuscript is well written, several modifications should be provided before publication.
Minor revisions:
Line 161: in the text, probably authors refer to Figure 2 and not Figure 3.
Line 172: in the text, probably authors refer to Figure 3 and not Figure 4.
Major revisions:
- Section 2.1. The authors performed a wide and specific work of selection for determining 65 possible AMPs, using different predictive programs and algorithms and 3D models were created with I-TASSER-MR server. However, authors choose only six peptides and it results difficult to understand why they considered only these. Did the authors select only peptides with α-helical structure after I-TASSER analysis?
As it is well explained in the introduction, although the most widespread AMPs are amphipathic α-helical peptides, these molecules present many forms. Indeed, the well-known human defensins possess β-sheet structures. In this case, maybe other sequences should be considered among these 65. For this reason, in my opinion this step should be better explained, even in the discussion or in the material and methods section.
- Section 2.4. The cytotoxic effect of AMPs was tested on McCoy cells and evaluated with two different techniques: fluorescent imaging and colorimetric assay. Since these two experimental approaches are well explained in the section of Material and Methods, authors could insert a figure showing the effects obtained with the most significant peptides (in each sense) and of the Melittin control.
In the figure S2, cell viability has been tested only on three peptides (Pept_1545, Pept_148 and Pept_84), why the authors did not test also the LHD assay on every peptide? The decision to analyze these derives from the fluorescent results? In this case, Pept_1303 showed a percentage higher than Pept_84.
Moreover, these experiments revealed as some peptides are significantly less cytotoxic compared with Melittin, used as control. Although, in the discussion, this AMP, deriving from bee venom, is described to possess a nonspecific cytotoxicity and haemolytic activity. In this case, why authors choose proper this molecule as control? If the final score is to discover new AMPs that could be used for the development of therapeutic agents, why they did not observe differences with other AMPs, such as the mentioned PAC-113 or dalbavancin, which are already used in this field and that not present cytotoxicity?
- Section 2.5. As before, a figure or images should be added, in which bacteria treated with propidium iodide are visible. Also in this case, the most significant conditions could be used.
Moreover, in these experiments, the effect of Pept_148 were evaluated during the kinetics of PI penetration (Figure 2), while no data are reported about the fluorescent result (Figure S3). Why the authors decided to not consider this peptide in the S3 graphs?
Why the same test has not been conducted on B. haemolyticus in order to compare AMPs effects with SEM results?
- Section 2.6. Why the authors decided to perform SEM analyses on the Pept_84, which not seems to show a great effect on subtilis in the PI assay and that possess high cytotoxicity, rather not considering the Pept_148?
Moreover, the AMP activity tested in the experiments with PI was calculated using the 2x MIC concentration and after a short period of time. Authors suggested a rapid mechanism of action for the antimicrobial peptides. Based on this experimental concentration and timing, why the authors choose 8 hours before observing SEM results and did not used the same concentration for compare the results?
Discussion
It seems that these six peptides derive from specific bacterial taxa present in the H. medicinalis microbiome (Aminobacter, Sphingobacteriia and Chitinophagia). Is there a correlation between these precise organisms and AMPs? Have these bacteria a precise role in the Hirudo microbiome? This point should be argument in the discussion.
Other important aspects about AMPs are both the positive surface charge and the hydrophobicity, as explained at line 228. Authors should add some information about these points (considering for example the Boman index), analysing the characteristics of the main peptides (or only of Pept_1545).
Author Response
We would like to thank the reviewer for their detailed comments and suggestions for the manuscript. We believe that the comments have identified important areas which required improvement. After completion of the suggested edits, the revised manuscript has benefitted from an improvement in the overall presentation and clarity. The detailed answers on issues as well as the modifications introduced into the revised version of the manuscript are given below. Original reviewer comments in boldface, responses in regular typeface.
In the present paper, the authors analyze the antimicrobial characteristics of six different bacterial peptides, selected after a specific in silico screening analysis conducted on Hirudo medicinalis metagenome. The MIC (minimum inhibitory concentration) of every chosen peptide have been calculated observing the effects on different Gram-negative and Gram-positive bacterial strains and the relative antibacterial role was evaluated by different techniques. Moreover, the cytotoxicity of these amino acid sequences has been tested on McCoy cells. Taken together, the results of these experiments highlight as these putative AMPs show a different antimicrobial activity and cytotoxicity.
In my opinion, although the manuscript is well written, several modifications should be provided before publication.
Minor revisions:
Line 161: in the text, probably authors refer to Figure 2 and not Figure 3.
Figure reference was corrected.
Line 172: in the text, probably authors refer to Figure 3 and not Figure 4.
Figure reference was corrected.
Major revisions:
- Section 2.1. The authors performed a wide and specific work of selection for determining 65 possible AMPs, using different predictive programs and algorithms and 3D models were created with I-TASSER-MR server. However, authors choose only six peptides and it results difficult to understand why they considered only these. Did the authors select only peptides with α-helical structure after I-TASSER analysis?
As it is well explained in the introduction, although the most widespread AMPs are amphipathic α-helical peptides, these molecules present many forms. Indeed, the well-known human defensins possess β-sheet structures. In this case, maybe other sequences should be considered among these 65. For this reason, in my opinion this step should be better explained, even in the discussion or in the material and methods section.
Peptides were chosen not by only their potential effects, but also on their secondary structure. From filtered 65 peptides, we chose 5 peptides with highest antimicrobial propensity scores which can exhibit α–helical structure and additionally 1 peptide with unstructured conformation and high propensity score based on the predictions.
The sentences “Overall, as potential AMPs, we identified 6 amino acid sequences (Table 1). Five of these peptides can adopt an α–helical structure under prediction. The leftover peptide is unstructured.” in the Result section were replaced by “According to the secondary structure prediction among filtered 65 peptides with the highest antimicrobial propensity scores the predicted secondary structures were only α–helical or unstructured conformations. From 65 peptides we chose 5 peptides with the highest antimicrobial propensity scores which can exhibit α–helical structure and additionally 1 peptide with unstructured conformation and high propensity score according to the predictions (Table 1).”
The sentence “As a result, we selected six putative α–helical AMPs.” in the Discussion section was corrected to “As a result, we selected six putative AMPs.”
- Section 2.4. The cytotoxic effect of AMPs was tested on McCoy cells and evaluated with two different techniques: fluorescent imaging and colorimetric assay. Since these two experimental approaches are well explained in the section of Material and Methods, authors could insert a figure showing the effects obtained with the most significant peptides (in each sense) and of the Melittin control.
In the figure S2, cell viability has been tested only on three peptides (Pept_1545, Pept_148 and Pept_84), why the authors did not test also the LHD assay on every peptide? The decision to analyze these derives from the fluorescent results? In this case, Pept_1303 showed a percentage higher than Pept_84.
Moreover, these experiments revealed as some peptides are significantly less cytotoxic compared with Melittin, used as control. Although, in the discussion, this AMP, deriving from bee venom, is described to possess a nonspecific cytotoxicity and haemolytic activity. In this case, why authors choose proper this molecule as control? If the final score is to discover new AMPs that could be used for the development of therapeutic agents, why they did not observe differences with other AMPs, such as the mentioned PAC-113 or dalbavancin, which are already used in this field and that not present cytotoxicity?
Due to the big amount of original data only the table with information about viability AMP–treated McCoy cells calculated from fluorescent images was presented in the manuscript. Figure S2 showing the effects obtained with the peptides and of the Melittin control were added in the Supplementary section.
Figure S2. Effect of AMPs on the survival of McCoy cells. Cells were treated for 24 h with peptides at a final concentration equal to 4× MIC. After staining with calcein AM and ethidium homodimer–1, samples were analysed by fluorescence microscopy. The peptide melittin was used as a positive control. The negative control corresponds to the McCoy cells incubated without peptide (DMEM).
Figure S3 was replaced with the figure showing results for all 6 peptides treatment during 24 hours.
Figure S3. Viability of AMP–treated McCoy cells measured by LHD assay. Cells were treated for 24 h with peptides at a final concentration equal to 4× MIC. Melittin was used as a negative control. The positive control corresponds to untreated cells (Control). The values are indicated as the mean ± SD (n = 3). Statistically significant differences between the control and experimental groups were determined by the non–parametric Kruskal–Wallis test, *p < 0.05, ***p < 0.005.
As positive control, we could use other AMPs with nontoxic effect, but in our case was used untreated cells that are more demonstrative as we suppose.
- Section 2.5. As before, a figure or images should be added, in which bacteria treated with propidium iodide are visible. Also in this case, the most significant conditions could be used.
Moreover, in these experiments, the effect of Pept_148 were evaluated during the kinetics of PI penetration (Figure 2), while no data are reported about the fluorescent result (Figure S3). Why the authors decided to not consider this peptide in the S3 graphs?
Why the same test has not been conducted on B. haemolyticus in order to compare AMPs effects with SEM results?
There are too many data to be shown in supplementary section that is why we decided to show only data in table that is more informative. Figure S4 showing the effects of the peptides on bacteria treated with propidium iodide was added in the Supplementary section.
Figure S4. Effect of AMPs on the survival of bacterial cells after peptide treatment. B. subtilis (top row) and E. coli (bottom row) were incubated with peptides at a final concentration equal to 2× MIC at 37 °C for 1 h. After staining with PI, samples were analyzed by fluorescence microscopy. The peptide melittin was used as a positive control. The negative control corresponds to the bacteria incubated without peptide (MHB). The size of each individual frame is 222 μm× 222 μm.
Pept_148 demonstrated low intensity of PI penetration that is why it is difficult to unambiguously defy the fluorescent signal from background and peptide. Due to this fact the results are controversial and didn’t show in the publication. Moreover, we tested two bacteria E. coli and B. subtilis, and peptide pept_148 didn’t exhibit antimicrobial activity against E. coli.
- haemolyticusis very pathogenic and, unfortunately, our laboratory doesn’t equip properly (with the fluorescent microscopes and spectrofluorometer in the specials boxes) for that kind of experiments. That’s the reason why these experiments were not conducted.
- Section 2.6. Why the authors decided to perform SEM analyses on the Pept_84, which not seems to show a great effect on subtilis in the PI assay and that possess high cytotoxicity, rather not considering the Pept_148?
Moreover, the AMP activity tested in the experiments with PI was calculated using the 2x MIC concentration and after a short period of time. Authors suggested a rapid mechanism of action for the antimicrobial peptides. Based on this experimental concentration and timing, why the authors choose 8 hours before observing SEM results and did not used the same concentration for compare the results?
As we examined the peptide effect on three bacteria by SEM we decided to investigate four peptides that affected all of them. Due to the preparation the SEM samples the concentration of peptides and bacteria were different from the experiments with PI to obtain sample with the cell monolayers. Moreover, in experiments with the kinetics of PI penetration, we used 2*MIC peptide concentration in order to perform experiments more quickly and avoid the possible effects of fluorescent dye fading.
Discussion
It seems that these six peptides derive from specific bacterial taxa present in the H. medicinalis microbiome (Aminobacter, Sphingobacteriia and Chitinophagia). Is there a correlation between these precise organisms and AMPs? Have these bacteria a precise role in the Hirudo microbiome? This point should be argument in the discussion.
Other important aspects about AMPs are both the positive surface charge and the hydrophobicity, as explained at line 228. Authors should add some information about these points (considering for example the Boman index), analysing the characteristics of the main peptides (or only of Pept_1545).
It is difficult to judge the role of these bacteria for the leech organism based only on metagenomic data; this requires additional experiments for a deeper understanding of the interaction between the host and its microbiota, unfortunately this is outside the scope of our study.
The corresponding discussion about specific bacterial taxa present in the H. medicinalis microbiome and their role in organism was added to the revised manuscript in the Discussion section:
“The selected peptides belong to the two phylums of Gram-negative bacteria Bacteroidetes (Sphingobacteriia and Chitinophagia) and Proteobacteria (Aminobacter), bacteria which are typically members of host-associated microbial communities. The literature data concern only investigations of metagenomic data and individual genomes from soil samples. There is little information on AMPs identified in these bacteria. At the moment only non-ribosomal antimicrobial peptides isolated from marine Proteobacteria has been known [35]. According to the existing observations and studies on the metagenomes of leeches of the genus Hirudo these taxa of microorganisms are common symbionts of leeches. The genus Aminobacter belongs to the order Rhizobiales, representatives of this order are components of the microbiota from the mucus and skin of the leech [36,37]. Representatives of bacteria of the class Sphingobacteriia and Chitinophagia are associated with metagenomes of the skin and excretory system of leeches [37,38]. For example, bacteria from the genus Niabella (Class Chitinophagia) can be found in leech cocoons, during the development of the leech, these bacteria colonize the bladder of the leech [39]. In some cases, representatives of these classes of bacteria were observed in insignificant quantities in various parts of the leech's digestive system [40, 41]. Thus, it can be assumed that these bacteria are endosymbionts and use antimicrobial peptides to compete successfully in the process of colonization of certain leech organs.”
Information about physical-chemical characteristics of AMPs was added in the Materials and Methods and Supplementary sections.
Table S2. The physical-chemical characteristics of AMPs from the H. medicinalis microbiome.
|
Peptide |
Amino acid sequence |
Charge |
Protein-binding potential (Boman index), kcal/mol |
Hydrophobic ratio, % |
|
pept_1303 |
IGRHFKRRNSIWGICWF |
+4 |
2.14 |
47 |
|
pept_148 |
VLIRGLIHMLRGG |
+2 |
0.05 |
53 |
|
pept_970 |
FVKILAKLVNYAKN |
+3 |
0.04 |
57 |
|
pept_1545 |
FLIGKAIKRKFCLRSVWNA |
+5 |
0.98 |
57 |
|
pept_352 |
KKGKSFKQLHIIVHLVKSWLRTILTHI |
+6 |
0.75 |
44 |
|
pept_84 |
IVKRFFRISYKLQSLKIIKGKRTFT |
+8 |
1.86 |
40 |
Analysis of physical-chemical characteristics of peptides demonstrated that all peptides have a positive net charge, the value of Boman index is varied from 0.04 to 2.14 and hydrophobic ratio is about 40% and higher. There is no reliable correlation between these parameters and antimicrobial action of peptides [Boman HG. Antibacterial peptides: basic facts and emerging concepts. J Intern Med. 2003;254(3):197-215. doi:10.1046/j.1365-2796.2003.01228.x]. We assume, there is no point in adding such information in the discussion.
The sentences “As a result, we chose 5 peptides with the highest antimicrobial propensity scores which can exhibit α–helical structure and additionally 1 peptide with unstructured conformation and high propensity score according to the predictions. The physicochemical properties of peptides were calculated using an APD3 server for the amino acid composition, hydrophobic ratio, Boman index, and molecular weight [4] (Table S2). “ were added in the Materials and Methods section.
The corresponding references were added to the revised version of the article.

Reviewer 2 Report
The work "The Hirudo Medicinalis Microbiome Is a Source of New Antimicrobial Peptides" by Grafskaia et al. presents in a clear fashion interesting results regarding antimicrobial activity of peptides whose sequences have been inferred from metagenomic study of Hirudo medicinalis microbiome.
I do not have any objections regarding presentation or interpretation of the results. However a few minor remarks have come to my mind.
- In materials and methods there is a significant gap regarding how the final 6 peptides were selected. In the result section there is a transition from 65 to 6 peptides without giving any rationale behind it. Please make it clear in the final manuscript.
- As the most active AMPs do not form alpha-helices in the examined conditions, what could be their potential mechanism of action? Is there a possibility they form alpha-helices during interaction with bacterial membranes of concrete species only? Wouldn't be relatively easy to obtain liposomes from the examined bacterial species and investigate such a possibility? Could authors deliver such an experiment for the investigated peptides?
- What could be a potential reason for the quite promising peptide 1545 to be highly effective towards S. haemolyticus but completely ineffective towards S. aureus? Could authors include in the manuscript any interpretation of this difference based on available research?
- Is it possible to suggest how should the peptide 1545 be modified in order to make it effective towards S. aureus? Would it be possible to include such a suggestion or a few in the discussion?
Author Response
We would like to thank the reviewer for detailed comments regarding the manuscript. We agree that dataset analysis is not totally clear and we implemented corresponding modifications. Responding on general suggestions we want to underline that our paper discusses the identification and functional analysis of the new antimicrobial peptides of Hirudo medicinalismicrobiome. Below you can find the detailed responses to the issues. Original reviewer comments in boldface, responses in regular typeface.
The work "The Hirudo Medicinalis Microbiome Is a Source of New Antimicrobial Peptides" by Grafskaia et al. presents in a clear fashion interesting results regarding antimicrobial activity of peptides whose sequences have been inferred from metagenomic study of Hirudo medicinalis microbiome.
I do not have any objections regarding presentation or interpretation of the results. However a few minor remarks have come to my mind.
- In materials and methods there is a significant gap regarding how the final 6 peptides were selected. In the result section there is a transition from 65 to 6 peptides without giving any rationale behind it. Please make it clear in the final manuscript.
The structures of AMPs are various but the majority of known AMPs form α–helix. Among filtered 65 peptides with the highest antimicrobial propensity scores the predicted secondary structures were only α–helical or unstructured conformations. According to our previous paper the highest AMP propensity calculated by predictors doesn’t guarantee the best antimicrobial action. That is why peptides were chosen not by only their potential effects but on secondary structure too. From 65 peptides we chose 5 peptides with the highest antimicrobial propensity scores which can exhibit α–helical structure and additionally 1 peptide with unstructured conformation and high propensity score according to the predictions.
The sentences “Overall, as potential AMPs, we identified 6 amino acid sequences (Table 1). Five of these peptides can adopt an α–helical structure under prediction. The leftover peptide is unstructured.” in the Result section were replaced by “According to the secondary structure prediction among filtered 65 peptides with the highest antimicrobial propensity scores the predicted secondary structures were only α–helical or unstructured conformations. From 65 peptides we chose 5 peptides with the highest antimicrobial propensity scores which can exhibit α–helical structure and additionally 1 peptide with unstructured conformation and high propensity score according to the predictions (Table 1).”
The sentence “As a result, we selected six putative α–helical AMPs.” in the Discussion section was corrected to “As a result, we selected six putative AMPs.”
- As the most active AMPs do not form alpha-helices in the examined conditions, what could be their potential mechanism of action? Is there a possibility they form alpha-helices during interaction with bacterial membranes of concrete species only? Wouldn't be relatively easy to obtain liposomes from the examined bacterial species and investigate such a possibility? Could authors deliver such an experiment for the investigated peptides?
Due to the mechanism of action is dependent on many factors as secondary structure, type of pathogens, environmental conditions and peptide concentration it is difficult to define the actual mechanism of AMP. On our findings, we only can assume that the mechanism of the most active AMP is likely non membranolytic but it must be proved by additional experiments. It is a fair point that the experiments with liposomes of different composition mimicking various bacteria could clarify the precise mechanism of action. In our previous study, we demonstrated that AMPs indeed form alpha-helices in the lipid environment [Grafskaia, E. N.; Nadezhdin, K. D.; Talyzina, I. A.; Polina, N. F.; Podgorny, O. V.; Pavlova, E. R.; Bashkirov, P. V.; Kharlampieva, D. D.; Bobrovsky, P. A.; Latsis, I. A.; Manuvera, V. A.; Babenko, V. V.; Trukhan, V. M.; Arseniev, A. S.; Klinov, D. V.; Lazarev, V. N. Medicinal leech antimicrobial peptides lacking toxicity represent a promising alternative strategy to combat antibiotic–resistant pathogens. Eur. J. Med. Chem. 2019, 180, 143–153. https://doi.org/10.1016/j.ejmech.2019.06.080.]. The AMPs interacted only with anionic lipids mimicking the bacteria but did not effect on zwitterionic vesicles which simulated eukaryotic cell membrane. Moreover, the secondary structure predictions of peptides didn’t take into account the environmental conditions, but that factor has a profound effect on the peptide conformation.
- What could be a potential reason for the quite promising peptide 1545 to be highly effective towards S. haemolyticus but completely ineffective towards S. aureus? Could authors include in the manuscript any interpretation of this difference based on available research?
We assume that the membrane composition of S. haemolyticus and S. aureus differ that results in different interaction with peptide and potential relevance to antibiotic response. Moreover, the considerable plasticity of S. aureus membrane lipid composition affects the response and resistance to various antimicrobial agents [Durham, D.R.; Kloos, W.E. Comparative study of the total cellular fatty acids of Staphylococcus species of human origin. Int. J. Syst. Bateriol. 1978, 28, 223–228, Tiwari, Kiran B et al. “Plasticity of Coagulase-Negative Staphylococcal Membrane Fatty Acid Composition and Implications for Responses to Antimicrobial Agents.” Antibiotics (Basel, Switzerland) vol. 9,5 214. 28 Apr. 2020, doi:10.3390/antibiotics9050214].
The sentences “We suppose that the difference between the response of S. haemolyticus and S. aureus to peptides dependent of membrane lipid composition of bacteria. The considerable membrane plasticity of S. aureus can provide antimicrobial resistance to peptides [43,44].” were added in the Discussion section.
- Is it possible to suggest how should the peptide 1545 be modified in order to make it effective towards S. aureus? Would it be possible to include such a suggestion or a few in the discussion?
After the precise investigation of Staphylococcus membrane composition, it can be possible to define proper amino acid substitution which probably improves the antimicrobial activity. On the other hand, this substitution can change the toxicity of the peptides that should be examined as well.
The corresponding references were added to the revised version of the article.
